# *Limosilactobacillus fermentum* CECT5716: Mechanisms and Therapeutic Insights

**DOI:** 10.3390/nu13031016

**Published:** 2021-03-21

**Authors:** María Jesús Rodríguez-Sojo, Antonio Jesús Ruiz-Malagón, María Elena Rodríguez-Cabezas, Julio Gálvez, Alba Rodríguez-Nogales

**Affiliations:** 1CIBER-EHD, Center for Biomedical Research (CIBM), Department of Pharmacology, University of Granada, 18071 Granada, Spain; mariajesus.rodriguez.sojo@gmail.com (M.J.R.-S.); a.jesus.ruiz14@gmail.com (A.J.R.-M.); merodri@ugr.es (M.E.R.-C.); albarn@ugr.es (A.R.-N.); 2Instituto de Investigación Biosanitaria de Granada (ibs.GRANADA), 18014 Granada, Spain; 3Servicio de Digestivo, Hospital Universitario Virgen de las Nieves, 18012 Granada, Spain

**Keywords:** probiotic, *Limosilactobacillus fermentum* CECT5716, immunomodulation, dysbiosis, mechanisms of action, gastrointestinal diseases, microbiota

## Abstract

Probiotics microorganisms exert their health-associated activities through some of the following general actions: competitive exclusion, enhancement of intestinal barrier function, production of bacteriocins, improvement of altered microbiota, and modulation of the immune response. Among them, *Limosilactobacillus fermentum* CECT5716 has become one of the most promising probiotics and it has been described to possess potential beneficial effects on inflammatory processes and immunological alterations. Different studies, preclinical and clinical trials, have evidenced its anti-inflammatory and immunomodulatory properties and elucidated the precise mechanisms of action involved in its beneficial effects. Therefore, the aim of this review is to provide an updated overview of the effect on host health, mechanisms, and future therapeutic approaches.

## 1. Introduction

The joint Food and Agriculture Organization (FAO) and World Health Organization (WHO) defined probiotics as “live microorganisms which when consumed in adequate amounts, confer a health effect on the host” [1]. Different characteristics are usually required to consider a microorganism as a probiotic, including: (i) must be taxonomically characterized, (ii) able to survive to the human intestinal environmental conditions, (iii) alive in sufficient numbers in the product at an efficacious dose throughout shelf life, (iv) supported by at least one positive human clinical trial conducted according to generally accepted scientific standards, and (v) safe for the intended use. Concerning the latter, most of the probiotics are categorized by Food and Drug Administration (FDA) as Generally Recognized as Safe (GRAS). 

The microorganisms mostly considered as probiotics belong to the *Lactobacillus* and *Bifidobacterium* genera, but also to other lactic acid bacteria, such as *Lactococcus spp.* and *Streptococcus thermophilus*. Other probiotic strains include the genera *Bacillus*, *Escherichia* (*E. coli* Nissle 1917), and *Propionibacterium* or yeasts like *Saccharomyces boulardii.*

Probiotics exert their health-associated activities through some of the following general actions (Figure 1):Competitive exclusion of pathogenic microorganisms. This occurs when one species of bacteria competes for receptor sites in the intestinal tract more actively than other species [2].Enhancement of intestinal barrier function. The intestinal barrier function plays an important role in the absorption of nutrients from food and, at the same time, prevents the access of potentially harmful bacteria to the human body [3]. When the gut barrier is disrupted, food antigens and pathogenic microorganisms can develop intestinal disorders, mainly associated with a local inflammatory response [4]. It has been proposed that probiotics maintain the epithelial barrier function, through increased expression of junction proteins or mucins, and promote intestinal epithelial cell activation in response to bacterial infection [5,6].Production of bacteriocins. These are antimicrobial peptides that prevent the proliferation of selected pathogens [7].Improvement of the altered microbiota composition. In normal conditions, the gut is colonized by a large number of microorganisms in balance, to provide energy and nutrition, maintain the intestinal immune homeostasis and protect the intestinal structure [8]. This balance is altered in many diseases, leading to a situation known as dysbiosis [9].Modulation of the immune response. Immunomodulation can be achieved by several mechanisms, including the modulation in the expression and/or production of anti- and pro-inflammatory cytokines [10] or increased production of immunoglobulins (Ig) [11].

In addition to these general actions exerted by the probiotics, it has been reported that other effects can also participate, which may be species- or even strain specific. However, the exact underlying mechanisms of action for each probiotic are still unclear, and the efficacy following their administration is quite different depending on the probiotic strain. Therefore, a deeper understanding of the mechanisms involved in the beneficial effects exerted by probiotics is especially relevant, and it should be considered that they must be characterized for each specific probiotic.

The aim of this review is to provide an overview of the current applications of the probiotic *Limosilactobacillus fermentum* CECT5716 and its potential use in different conditions, based on its specific mechanisms reported in different preclinical and clinical studies. 

## 2. Limosilactobacillus fermentum

Lactic acid bacteria (LAB) belong to the phylum *Firmicutes*, class *Bacilli*, order *Lactobacillales.* They are considered to play important roles in food production, nutritional supplementation, agriculture, as well as in veterinary and human medicine [12]. LAB are Gram-positive bacteria, generally without catalase activity [13] that are able to produce lactic acid as the main end-product after carbohydrate fermentation. Different genera are considered in this LAB group: *Aerococcus, Carnobacterium, Enterococcus, Lactobacillus, Lactococcus, Leuconostoc, Oenococcus, Pediococcus, Streptococcus, Tetragenococcus, Vagococcus,* and *Weissella* [14]. LAB are present in different ecological niches, and many studies have established differences when considering their genetic and physiology [15]. The genus *Lactobacillus* is certainly the most studied genus in the LAB group, and more than 240 species have been reported to be included [16], which can be found at different localizations in the human body, including the gastrointestinal tract, as well as the urinary and genital systems. As commented above, this genus is considered one of the most important representative groups of probiotics [17], and particularly, *Lactobacillus fermentum* has become one of the most promising probiotics. In fact, it is used as a standard reference species in comparative studies with other probiotics, due to their beneficial health properties [17].

Recently, whole genome studies have been performed, and the taxonomy of *Lactobacillaceae* has been newly evaluated. Thus, the previous name, *Lactobacillus fermentum*, has been changed by *Limosilactobacillus fermentum* [18].

*L. fermentum* is a species with many strains isolated from different environments, including fermenting plant materials [19], dairy products [20], bread [21], naturally fermented sausages [22], breast milk [23], saliva [24], and human feces [25]. Remarkably, several *L. fermentum* strains have been described to possess promising beneficial effects both in preclinical studies (in vitro and in vivo models) and in human trials, which, in fact, have resulted in the development of different probiotic preparations for medical application and food preservation processes [26]. Actually, it has been suggested a potential role of *L. fermentum* in inflammatory-related diseases including intestinal inflammation [27], respiratory tract infections [28] and hepatic injury [29] (Table 1 and Table 2).

As expected, the mechanisms involved in the proposed beneficial effects reported for these probiotics include the general mechanisms described above [30,31]. Among these, the immunomodulatory properties have been proposed to have a key role in many strains of *L. fermentum*, since they are able to interact with immune cells, like macrophages and dendritic cells, as well as to regulate the synthesis and release of different cytokines [32,33]. In addition, *L. fermentum* strains have been proposed to exert bacteriostatic effects against a variety of pathogenic bacteria and fungi, including *Staphylococcus aureus* [34], *Candida albicans* [35], *Helicobacter pylori* [36], *Campylobacter jejuni* [37], and *Aspergillus parasiticus* [38], derived from their ability to produce organic acids (primarily lactic and acetic acids) and/or antimicrobial peptides [26,39]. Furthermore, it has been reported that some *L. fermentum* strains possess a complete glutathione-associated system, including the synthesis, transport, uptake, and redox cycling of this antioxidant peptide [40,41], thus providing protection against oxidative stress (Table 1 and Table 2).

However, of all strains identified from *L. fermentum*, *L. fermentum* CECT5716 is one of the probiotics with more potential. Different studies have reported its possible beneficial effects in different pathologies [57,58]. However, the precise mechanisms underlying remain unknown. Therefore, it is still necessary to conduct more investigations to identify its mechanisms of action and possible interactions with the host. Here, we will summarize and provide updated information on its effect on host health, mechanisms, and therapeutic insights. 

### 2.1. Limosilactobacillus fermentum CECT5716

*L. fermentum* CECT5716 is a probiotic strain initially isolated from the human breast milk of healthy mothers, and for over 15 years, it has been included in nutrition supplements and fermented milk products [23]. The application of whole-genome shotgun strategies provided the identification of its genome using *L. fermentum* IFO 3956 as reference. Both strains are highly similar, with the exception of 16 protein encoding genes that are not present in IFO 3965 [59]. Thus, the genome of *L. fermentum* CECT5716 is composed of 2100449 bp and contains 1109 protein encoding genes, 54 tRNA encoding genes, and 20 rRNA encoding genes. It is a circular chromosome with a CG content of 51.49%, with no plasmid, and includes putative enzymes with an important role in the metabolism of purines (allantoinase, guanosine monophosphate (GMP) oxidoreductase, GMP synthase), amino acids (serine-pyruvate transaminase, 3 glutamate synthases), lipids (acyltransferase), and carbohydrates (mannose-6-phosphate isomerase) (GenBank/EMBL under accession no. CP002033) [59].

#### 2.1.1. Preclinical Studies

*L. fermentum* CECT5716 has been assayed in in vitro studies as well as in different preclinical models such as experimental colitis, metabolic syndrome, lupus, and lactancy, among others. Thus, it has shown potential benefits in health and disease, and the investigations have provided precious information about the mechanisms involved (Table 3 and Table 4). In addition, these studies have confirmed the safety of its intake, thus discarding any toxicity on the host.

In vitro studies

Epithelial cells.

CMT-93 cell line.

This cell line is derived from mouse rectum, and it is used as a model of intestinal epithelial barrier to study the effects of different pharmacological agents on gut permeability [60], intestinal inflammation [58] or colorectal cancer [61]. Therefore, CMT-93 cells are a helpful tool to validate the effect or the mechanism of action of probiotics. Different studies have been conducted using this cell line to evaluate the effectiveness of *L. fermentum* CECT5716 administration. The incubation of lipopolysaccharide (LPS)-stimulated CMT-93 cells with *L. fermentum* CECT5716 has shown a significant reduction in the expression of different proinflammatory cytokines, including *Il-6* and *Tnf-α*, as well as an increase in the expression of mucins, which can account to ameliorate the compromised gut barrier integrity in an inflammatory environment [58]. Furthermore, *L. fermentum* CECT5716 and CMT-93 co-culture revealed that the probiotic is able to restore the altered expression of different miRNAs, including *miRNA-150, miRNA-155,* and *miRNA-375* [58], described as important regulators of intestinal inflammation, fine-tuning the immune system and the mucosal barrier functions (Table 3).

b.Caco-2 cell line.

Caco-2 cell line was isolated from a human colorectal adenocarcinoma [62] and has become one of the most widely used models for intestinal absorption, bioavailability, and inflammatory studies [63,64]. Investigations performed in this cell line have supported the potential of *L. fermentum* CECT5716 as a therapeutic tool to manage intestinal inflammation. In fact, pretreatment of *L. fermentum* CECT5716 to stimulated Caco-2 cells has shown a significant decrease in nitric oxide (NO), IL-8, and IL-1β in comparison with control cells. Moreover, it was associated with a reduced phosphorylation of the Mitogen-activated protein kinase (MAPK) p42/44 extracellular signal-regulated kinase (ERK) and p38 when compared to stimulated cells without probiotic [50]. Surprisingly, Rodriguez-Nogales et al. (2015) [65] demonstrated that *L. fermentum* CECT5716 viability was not necessary for its immunomodulatory properties, reporting that this strain, live or dead, inhibited IL-8 production by Caco-2 stimulated with IL-1β. This probiotic was also able to inhibit MAPK activity showing a reduction in the phosphorylation of the MAPK p42/44 ERK and p38 [65] (Table 3). These in vitro studies suggest that *L. fermentum* CECT5716 exerts immunomodulatory properties in epithelial intestinal cells, which might have a contributory role in the anti-inflammatory effect at the intestinal level (Table 3).

2.Macrophages

The immunomodulatory effects described on animal models and epithelial cells have been also reported in different antigen-presenting cells.

Bone marrow-derived macrophages (BMDM).

BMDM are primary cells and in contrast to macrophage cell lines, they are mature with biological function and properties of macrophages differentiated from monocytes. BMDMs have been used for genetic screening (RNAi), drug screening, functional studies, host–pathogen interaction studies, and many other areas of investigation [66,67,68,69,70]. Studies performed using this cell model have shown that the probiotic *L. fermentum* CECT5716 downregulated the expression of pro-inflammatory mediators such as *Tnf-a*, *Il-6,* and *iNos*. Moreover, it was also able to reduce TNF-α and IL-1β production on an inflammatory phenotype of BMDM stimulated with LPS (LPS-stimulated BMDM cells) [71]. Recently, it has also shown that the probiotic is able to restore the miRNA expression in LPS-stimulated BMDM cells. Specifically, the probiotic was able to modulate the altered expression of *miRNA-150, miRNA-155,* and *miRNA-375*. Theses miRNAs are associated with mucosal leukocyte infiltration [72], induction of the macrophage activation and antigen presentation by dendritic cells [73], and with the differentiation of goblet cells and the inflammatory response of macrophages [74,75], respectively. Interestingly, all these miRNAs are upregulated in IBD patients [76] and the probiotic was able to restore their expression in LPS-stimulated BMDM [58] (Table 3).

b.RAW 264.7 cells.

RAW 264.7 is a cell line derived from murine monocytes. RAW 264.7 cell line was established from a tumor induced by the leukemia virus [77]. This cell line presents macrophage properties [62], and it is used to study the anti-inflammatory effects of drugs and/or natural extracts [78,79]. Consequently, the immunomodulatory effect of *L. fermentum* CECT5716 has been also evaluated in this macrophage cell line. The pretreatment of these cells with *L. fermentum* CECT5716 resulted in a significant reduction in some mediators of inflammation [80]. Exactly, *L. fermentum* CECT5716 reduced IL-1β and NO production when compared with those cells untreated and LPS-stimulated [65]. Similarly, the probiotic, dead, was also capable of reducing IL-1β and NO production when compared with those cells untreated and stimulated with LPS [65] (Table 3).

c.Peripheral blood mononuclear cells.

Peripheral blood mononuclear cells (PBMCs) are extracted from whole blood. The PBMCs are separated in a layer using Ficoll. This fraction of polymorphonuclear cells is constituted by lymphocytes (T cells, B cells, and natural killer (NK) cells) and monocytes. PBMCs are used as a powerful tool to screen compounds with immunomodulatory properties and determine their mechanism of action [81,82]. A study performed in 2010 evaluated the potential activity of *L. fermentum* CECT5716 to modulate both human PBCMs activation and cytokine profile. The results showed that the incubation of *L. fermentum* CECT5716 produced activation of both NK cells (CD8^+^ NK subset) and Treg cells (CD4^+^CD25^+^Foxp3^+^). Moreover, the probiotic was able to induce different cytokines including Il-10, TNF-ɑ, MIP-1β, IL-1β, IL-1ɑ, IL-18, and IFN-γ implicated in the inflammatory process [83]. Thus, this study demonstrated that *L. fermentum* CECT5716 enhanced natural and acquired immune responses, as evidenced by the activation of NK and T cell subsets and the expansion of Treg cells, as well as induced a broad array of cytokines [83] (Table 3).

The effects shown by this probiotic strain in in vitro studies have been confirmed and supported by different preclinical studies. 

Animal models.

Models of experimental colitis.

These models resemble human inflammatory bowel disease (IBD). It is a medical term that mainly includes two different clinical states, Crohn’s disease (CD) and ulcerative colitis (UC), and refers to specific chronic inflammatory states in the gastrointestinal tract. Its etiology is complex and not fully identified, involving interaction between genes, such as NOD2 and ATG16L1, and a number of functional abnormalities, like imbalance between pro-inflammatory, and anti-inflammatory T cell responses [58], impaired epithelial barrier function and gut microbiota dysbiosis [58]. Connecting the heritability and the environmental influence, unique microRNA (miRNA) expression profile signatures have been associated with IBD pathogenesis. In this regard, both endogenous gut miRNAs and exogenous miRNAs from diet could actively modulate microbial colonization and intestinal immunity [84]. Thus, gut microbiota in IBD patients differs from healthy individuals, as it is characterized by a reduced diversity, increased abundance of *Enterobacteriaceae* and reduced abundance of *Firmicutes* and *Bacteroidetes* phyla [85]. As well as gut microbiota is key for the maintenance of gut homeostasis, it could also promote chronic inflammation and the development of IBD [86,87,88] (Table 4).

In this regard, IBD patients display a modified balance between the T helper (Th)17/T regulatory (Treg) and the Th1, Th2, and Treg populations [89], having been proposed that a downregulation of Treg together with an increase in Th17 cells participate in the loss of tolerance for microbiota, that triggers inflammation [90]. Additionally, since the gut microbiota produces host-beneficial substances, the changes observed in the gut microbiota of IBD patients are associated with decreased levels of short-chain fatty acids (SCFAs), which have anti-inflammatory properties and are one of the energy sources for colon cells [91] (Table 4).

Despite the complexity of IBD, different experimental models are available, being the dextran sulfate sodium (DSS), 2,4,6-trinitrobenzene sulfonic acid (TNBS), or dinitrobenzene sulfonic acid (DNBS), in rodents, the most used. They have been very useful for understanding the pathophysiology of these conditions, as well as for the development of new therapeutic approaches and the elucidation of their mechanisms of action [92]. *L. fermentum* CECT5716 has been evaluated in these models, obtaining promising results that suggest a potential application against intestinal inflammation (Table 4).

TNBS-induced colitis in rats.

This model consists of the intra-rectally administration of the hapten TNBS diluted in ethanol [93]. The intestinal inflammation initially results from ethanol-induced damage to intestinal epithelial cells, which increases permeability, microbial penetration into the mucosa, and haptenization of host proteins by the TNBS, all leading to infiltration of neutrophils, macrophages, and Th1 T lymphocytes into the damaged mucosa [93]. Therefore, this model is characterized by a dense colonic tissue infiltration by CD4 T cells and the secretion of various potent pro-inflammatory cytokines [94]. Preventive studies performed in TNBS-induced colitis in rats showed that oral administration of *L. fermentum* CECT5716 ameliorated the inflammatory response evidenced by downregulation of pro-inflammatory mediators such as tumor necrosis factor (*Tnf*)-α, interleukin (*Il*)-6, and inducible nitric oxide (*iNos*) [95,96]. The probiotic administration also reduced the histological damage score, as well as the myeloperoxidase (MPO) activity [95,96], common marker of neutrophil infiltration to evaluate the tissue damage in IBD [97]. Moreover, the probiotic supplementation was also able to increase the production of SCFAs in comparison with untreated colitic rats. Additionally, *L. fermentum* CECT5716 pretreatment counteracted colonic glutathione (GSH) depletion, which is considered a crucial event for colonic damage [98]. GSH is often recognized as a marker of oxidative stress and thus, as a consequence of its antioxidant properties, scavenging of both oxygen and nitrogen reactive species [95]. The preventive effects of *L. fermentum* CECT5716 may be due to both early immune stimulation and immune regulatory properties, as well to its antioxidant abilities [95,96]. Additionally, a curative study carried out with *L. fermentum* CECT5716 in the same TNBS model in rats showed its effect in the late phase of colitis, being able to decrease leukotriene B4 (LTB4) production, *Il-6* expression, and Toll-like receptor (TLR)/MyD88 function [95]. Based on these results, it can be stated that *L. fermentum* CECT5716 attenuates TNBS colitis, maybe due to its antioxidant capacities, and effectively accelerates colitis recovery by enhancing TLR function (Table 4). 

These results confirmed that *L. fermentum* CECT5716 exhibits one the important features of potential probiotic candidates, i.e., the capacity to modulate the immune response of the host, which clearly contributes to its intestinal anti-inflammatory effect. However, it is interesting to note that these beneficial effects are not exclusively dependent on the probiotic viability, since both live and dead probiotic ameliorated the production of some of the mediators involved in the colonic inflammation induced by TNBS in rats [65] (Table 4).

b.DNBS-induced colitis in mice.

This experimental model of colitis also employs rectal instillation of the mucosal sensitizing agent DNBS. Similarly to TNBS, the intestinal inflammation is characterized by an increased permeability, microbial penetration into the mucosa, infiltration of neutrophils, macrophages, and Th1 T lymphocytes [93]. *L. fermentum* CECT5716 has also demonstrated anti-inflammatory activity in this experimental model of mouse colitis. Its administration prevented the DNBS-induced inflammation by ameliorating weight loss and the incidence of diarrhea when compared with the untreated colitis group. Moreover, the treatment with *L. fermentum* CECT5716 increased the survival rates due to a reduction in the inflammatory process that it was manifested by the downregulation of several pro-inflammatory mediators such as *Tnf*-α, *Il-1β, iNos,* and metalloproteinase (*Mmp)-9* as well as amelioration of intestinal epithelial integrity showed by an increment of mucin (*Muc*)*-3* and *Occludin* [58]. The probiotic was also capable of restoring the levels of *miRNA-155* and *miRNA-223* both of them involved in the colitic process [99]. Moreover, the probiotic treatment ameliorated gut microbiota composition increasing microbial diversity and the *Firmicutes*/*Bacteroidetes* ratio, which is reduced in the DNBS-untreated group [58] (Table 4).

c.DSS-induced colitis in mice.

This model is based on the administration of DSS in drinking water resulting in damage to the intestinal epithelium. This causes a dissemination of the luminal contents into underlying tissue triggering an inflammatory response. It has also highlighted the intestinal anti-inflammatory effect of *L. fermentum* CECT5716 in this model. In fact, it has been confirmed that the preventative administration of *L. fermentum* CECT5716 exerted intestinal anti-inflammatory effects and ameliorated dysbiosis. Thus, the treatment with *L. fermentum* CECT5716 reshaped the gut microbiota, increasing microbial diversity and restoring the *Firmicutes*/*Bacteroidetes* ratio [27] (Table 4). 

In summary, all the preclinical studies performed to evaluate the effect of *L. fermentum* CECT5716 on IBD confirm its immunomodulatory activity and intestinal anti-inflammatory properties. Among the mechanisms involved, it is interesting to highlight the positive impact on the innate immune response, preserving the intestinal barrier integrity and decreasing pro-inflammatory cytokines production, and on the adaptive immune response, modulating the expression of Th1-, Th17-, and Treg-related cytokines. Moreover, it has been demonstrated the ability of *L. fermentum* CECT5716 to restore intestinal microbiota composition. Furthermore, it has been suggested that all these actions could be achieved at a post-transcriptional level by modifying the expression of some miRNAs, which is altered in these intestinal inflammatory conditions (Table 4).

Metabolic syndrome.

Metabolic syndrome is a cluster of several disorders that include hypertension, hyperglycemia, and hyperlipidemia. Obesity is the common point of these alterations that increase the risk of type-2 diabetes (T2D), cardiovascular disease, and cancer [100] (Table 4). Moreover, obese patients have been reported a subclinical chronic inflammation, called metaflammation [101] and associated with the secretion of pro-inflammatory mediators, such as IL-6 and TNF-α. Thus, it promotes the recruitment of macrophages to adipose tissues and contributes to the metabolic dysfunctions and obesity-related diseases in these patients [101] (Table 4). This condition is developed as a consequence of an energy imbalance due to an excessive energy intake and low expenditure, leading to an abnormal accumulation of lipids in metabolic tissues, mainly adipose tissue and liver [102,103]. Moreover, many studies have pointed out the role of gut microbiota in the development of metabolic syndrome [104]. It has been reported that an altered gut microbiota together with an increased gut permeability promote bacterial endotoxins translocation into the systemic circulation, which also contributes to the metaflammation [105]. In this sense, several natural compounds such as probiotics have been assayed for the treatment of metabolic syndrome [106,107] (Table 4). Among them, *L. fermentum* CECT5716 has demonstrated to be a promising candidate to prevent these metabolic dysfunctions. Hypertension is among the most prevalent risk factors for cardiovascular events such as stroke and myocardial infarction. Up to present date, an ever-increasing number of studies have shown a link between gut microbial signatures and hypertension [108,109]. Thus, a study performed in spontaneously hypertensive rats (SHR) revealed the potential cardiovascular effects of *L. fermentum* CECT5716 in genetic hypertension [110,111]. The study showed that chronic treatment with *L. fermentum* CECT5716 prevented both gut dysbiosis (reduced *Firmicutes*/*Bacteroidetes* ratio, increased butyrate-producing bacteria) and blood pressure increase in SHR. In addition, the treatment restored the Th17/Treg balance in mesenteric lymph nodes, and normalized endotoxemia, as well as prevented the impairment of endothelium-dependent relaxation to acetylcholine, as a result of reduced nicotinamide adenine dinucleotide phosphate (NADPH) oxidase-driven reactive oxygen species (ROS) production. The probiotic was also able to prevent the vascular oxidative stress and endothelial dysfunction mediated by the vascular LPS/TLR4 pathway [110,111] (Table 4). Besides, it has been demonstrated that a synbiotic composed of *L. fermentum* CECT5716 and fructooligosaccharides prevented the development of liver steatosis and inflammatory status in rats fed a high fructose diet. These effects were associated with changes in the microbiota, restoring dysbiosis occurred in non-treated rats. The synbiotic also ameliorated the glucidic profile by reducing the plasma levels of glucose and insulin, thus improving homeostasis model assessment of insulin resistance (HOMA-IR) index, which is used to determine the severity of insulin resistance and prevented the hypertriglyceridemia and hyperleptinemia that occurred in rats fed the high fructose diet [112] (Table 4). These findings reveal that this probiotic might be clinically useful in cardiovascular conditions, particularly considering that high fructose intake has been related to metabolic syndrome in humans. Supporting this, it has also been shown that *L. fermentum* CECT5716 prevents vascular oxidative stress and gut dysbiosis in rats with hypertension induced by chronic NO blockade [113] (Table 4). In this study, the authors showed that the chronic *L. fermentum* CECT5716 treatment reduced early events involved in atherosclerosis development, such as vascular oxidative stress and pro-inflammatory status, as a result of prevention of gut dysbiosis and immune changes in mesenteric lymph nodes, but not hypertension, confirming the critical role of NO in the antihypertensive effects of *L. fermentum* CECT5716 in genetic hypertension [113] (Table 4). 

Recently, *L. fermentum* CECT5716 has been evaluated in a model of high-fat diet (HFD)-induced obesity in mice [114] (Table 4). This is very interesting since obesity is one of the main features of metabolic syndrome. The treatment with *L. fermentum* CECT5716 to HFD-fed mice improved HFD-induced obesity, reducing body weight gain, which was associated with an amelioration of glucose and lipid metabolism. The treatment did not significantly modify energy intake, thus discarding any anorexigenic effect, but it showed anti-inflammatory and immunomodulatory properties, as seen before in different experimental models of intestinal inflammation [27,65] (Table 4). Indeed, *L. fermentum* downregulated the expression of pro-inflammatory mediators in HFD-fed mice as well as upregulated the expression of key transcription factors that control adiponectin such as *Ppar-α* [115]. As in human obesity, this was associated with an improvement in glucose and lipid metabolism, most probably derived from the amelioration of the obesity-associated insulin resistance, as shown by the impact of the probiotic treatment on HOMA-IR values. Furthermore, *L. fermentum* CECT5716 treatment ameliorated leptin resistance in liver and adipose tissue as well as restored adiponectin expression in fat contents, which has been previously reported for other probiotics, like different strains of *L. plantarum* [116,117]. Obesity-associated insulin resistance is also characterized by impairment of intracellular glucose uptake that is mediated by the insulin-dependent receptor GLUT-4 [103], whose expression is reduced when insulin resistance appears [118]. *L. fermentum* CECT5716 also improved insulin resistance, which was associated with an increased expression of *Glut-4*, ameliorating glycemic levels and glucose utilization by target tissues [119]. As commented before, the obesity-associated inflammatory state is closely related to the development of cardiovascular disease and endothelial dysfunction, in part, mediated by leptin [120] (Table 4). Interestingly, *L. fermentum* CECT5716 treatment reduced vascular expression of the pro-inflammatory cytokines *Tnf-α* and *Il-1β* in obese mice, as well as inhibited the increased NADPH activity in the aortic tissue. This suggests a reduction in ROS production and a higher NO bioavailability, which could promote the restoration of the impaired endothelium-dependent relaxation to acetylcholine [114]. It is well described that the gut plays an important role in the pathogenesis of obesity. In fact, there is a defect in the intestinal barrier function that leads to increased gut permeability [121] (Table 4). Therefore, bacterial components, like LPS, could reach systemic circulation and provoke metabolic endotoxemia [122]. It has been confirmed these observations, since obese mice displayed reduced expression of the colonic markers of epithelial integrity as well as increased LPS plasma levels and upregulated expression of *Tlr-4* in liver, fat, and aorta [114]. Importantly, *L. fermentum* CECT5716 significantly increased the colonic expression of the different markers involved in gut integrity in obese mice, thus restoring the intestinal barrier function and preventing bacterial components translocation, since it reduced LPS plasma levels and downregulated *Tlr-4* expression [114]. In addition, it is well established that gut microbiota composition is altered in obesity [123], consisting of an enrichment in *Firmicutes* as well as a reduction in *Bacteroidetes*, both in humans [124] and mice [125]. Increased *Firmicutes*/*Bacteriodetes* ratio has been associated with a more efficient hydrolysis of non-digestible polysaccharides in the intestinal lumen, so obese individuals extract more calories and fat from food than lean ones [126]. It has also been demonstrated that *L. fermentum* CECT5716 treatment was able to modulate gut microbiota composition, restoring the main bacteria phyla to the normal values observed in control diet-fed mice [114]. This amelioration of obesity-associated dysbiosis could be associated with the reduction in energy assimilation and potentially contribute to the beneficial effects observed [114]. Collectively, the probiotic was able to exert anti-obesity effects, maybe related to its anti-inflammatory properties and amelioration of endothelial dysfunction and gut dysbiosis [114]. This study, for the first time, shows the ability of this probiotic to ameliorate experimental obesity through microbiome modulation, affecting different bacteria that have been reported to play a key role in the pathogenesis of obesity [114] (Table 4).

Taking into account all results on preclinical studies in metabolic syndrome, this probiotic represents a very interesting approach on the prevention and therapy of metabolic dysfunctions as well as they indicate a potential use of *L. fermentum* CECT5716 in clinical practice.

Systemic Lupus Erythematosus.

Systemic lupus erythematosus (SLE) is a chronic autoimmune disease associated with an increased risk of renal and cardiovascular disease development. SLE is characterized by an alteration of the immune system, in which abnormally functioning B lymphocytes promote an exacerbated production of autoantibodies that trigger the formation and deposition of immune complexes, which in turn damage many organs and tissues [127]. The etiopathogenesis is still not fully understood, however, it is widely recognized that SLE is the consequence of the effects of environmental factors in genetically predisposed individuals, leading to the disruption of self-tolerance and to the activation/increase in innate immune cells and autoreactive lymphocytes [128]. Although the mechanism of the disease involves both genetic and environmental factors, it has been found that the microbiota is also a key player [129]. The altered gut microbiota composition in SLE patients is associated with an imbalance in the proportions of Th17 and Treg cells [130]. Considering all these facts, the use of probiotics with immunomodulatory properties is widely justified for the treatment of this disease. In this scenario, it has been reported that *L. fermentum* CECT5716 exerted beneficial effects in an experimental model of SLE in mice, by ameliorating the cardiovascular complications associated with this disease [57] (Table 4). These beneficial effects were due to the ability of the probiotic to prevent intestinal dysbiosis occurred in experimental SLE. The administration of *L. fermentum* CECT5716 in mice with SLE restored the *Firmicutes*/*Bacteroidetes* ratio and increased the levels of bacterial species of beneficial genus such as *Bifidobacterium* and *Parabacteroides*. This restoration of the microbiota could be responsible of the immunomodulatory activity of this probiotic that reduced the percentage of B cells, as well as the levels of T reg (CD4^+^/FoxP3^+^) cells, Th17 (CD4^+^/IL-17a^+^) cells, and Th1 (CD4^+^/IFN-γ^+^) cells, which were increased in the mesenteric lymph nodes from SLE mice (Table 4). Moreover, the treatment with *L. fermentum* CECT5716 significantly reduced the expression of pro-inflammatory cytokines such as *Il-17a, Ifn-γ,* and *Il-21* in plasma. The probiotic also ameliorated the intestinal integrity and, together with the beneficial effects shown in the microbiota profile, resulted in a reduction in LPS absorption in treated mice. The endothelial dysfunction that occurred in SLE mice was reversed with the *L. fermentum* CECT5716 treatment, being associated with a reduced NADPH oxidase activity and normalization of the NO production by the endothelial nitric oxide synthase (eNOS) [57] (Table 4). The effect of *L. fermentum* CECT5716 on endothelial dysfunction was corroborated in previous studies, in which this probiotic was evaluated in the tacrolimus-induced hypertension in mice [131]. The tacrolimus is a macrolide used for maintenance of immunosuppression in organ transplant recipients that has been shown to affect blood pressure and immunologic memory. *L. fermentum* CECT5716 treatment ameliorated the hypertension induced by tacrolimus and prevented the endothelial dysfunction occurring in this experimental group. These effects were due to the probiotic ability of suppressing the release of endothelial-derived vasoconstrictor factors and normalizing the expression of cyclooxygenases (*Cox*)-*1* and *Cox-2*. The administration of *L. fermentum* CECT5716 reduced the vascular oxidative stress by a reduction in the activity of NADPH oxidase. It was also reported an amelioration of the immune response preventing the altered T-cell polarization induced by tacrolimus and restoring the plasma levels of IL-10 and IL-7. The activity demonstrated by the probiotic could be associated with the beneficial effects exerted on gut microbiota, which was altered by the tacrolimus administration. In fact, *L. fermentum* CECT5716 restored the ratio *Firmicutes*/*Bacteroidetes* that was increased in tacrolimus-administered mice [131] (Table 4). The preventative effect of *L. fermentum* CECT5716 on the development of endothelial dysfunction and high blood pressure in SLE genetic mice (NZBWF1) has been also evaluated, and the results obtained revealed that *L. fermentum* CECT5716 protected the kidney damage and reduces systolic blood pressure in these mice. These protective effects exerted by the probiotic were associated with a decrease in renal oxidative stress and inflammation, derived from the reduced plasma levels of autoantibodies and LPS [132].

All these findings support the use of *L. fermentum* CECT5716 in prophylaxis and/or therapy of the complications associated with SLE. These beneficial effects are attributed to the ability of the probiotic to modulate the gut microbiota and the immune response, reducing anti-double-stranded DNA (dsDNA) and the endotoxemia characteristic of SLE patients. 

Pregnancy and lactation.

Although these conditions are not considered as a disease, they comprise important physiological adaptations that affect the nutritional needs, the microbiota of the mother and, therefore the acquisition of the microbiota offspring. All of them can influence not only the mother’s health, but the correct development and growth of the newborn. In these periods, the immune system also undergoes alterations to achieve tolerance towards and adequate support for the fetus and for the lactation process [133]. Pregnancy and the increased metabolic demands during parturition and lactation result in increased production of ROS, leading to reductions in neutrophil function, antibody responses, and cytokine production by immune cells [134] (Table 4). Recently, an experimental model using rats during pregnancy and lactation has demonstrated the immunomodulatory ability of the *L. fermentum* CECT5716 treatment [133]. It has been shown that the treatment with the probiotic reduces the levels of cytotoxic T cells (CD8^+^ TCRαβ^+^ and TCRγδ^+^) in mesenteric lymph nodes of dams rats. Moreover, *L. fermentum* CECT5716 was able to induce changes on the fatty acid profile in plasma of dams and pups rats producing an increase in γ-linolenic acid and a reduction in total saturated fatty acids in dams [135]. These findings suggest that *L. fermentum* CECT5716 treatment can be a promising strategy on pregnancy and lactation (Table 4).

#### 2.1.2. Human Trials

Clinical studies are crucial to really establish the potential role of probiotics in human health and disease, thus providing evidence to support their use as a nutraceutical or functional food. Although the preclinical studies have shown promising results, the available clinical research focused on *L. fermentum* CECT5716 is scarce, and only a few human studies have been reported. The beneficial effects of *L. fermentum* CECT5716 have been validated on different conditions in which an altered immune response occurs, including mastitis and infections, mainly in infants (Table 5).

Mastitis

Mastitis is an inflammation of breast tissue that mainly occurs during lactation, most probably derived from an infection status, which has been associated with dysbiosis in the breast milk microbiota. The mastitis-associated dysbiosis is characterized by an increase in the proliferation of some bacterial species, mainly from *Staphylococcus* genus, together with the disappearance of other species, like lactobacilli or lactococci [136]. The administration of probiotics has been proposed as a strategy to restore this dysbiotic status and alleviate mastitis. Accordingly, and based on the immunomodulatory and anti-infectious properties reported for *L. fermentum* CECT5716, this probiotic could be a therapeutic approach of great value in this condition. In fact, different randomized double-blinded controlled trials have reported that the consumption of *L. fermentum* CECT5716 (at doses ranging from 3 × 10^9^ to 9 × 10^9^ CFU/day) prevented lactational mastitis and/or improved some of the inflammation-associated symptoms, including pain [137,138] (Table 5). In addition, the levels of the chemokine IL-8, which is considered as an indicator of mastitis, was also significantly lower in the breast milk of women treated with the probiotic. Of note, the beneficial effects of the probiotic against mastitis were associated with an 80% reduction in *Staphylococcus* counts in breastmilk, thus ameliorating the dysbiosis that characterizes this condition [138] (Table 5). Moreover, when the effect of *L. fermentum* CECT5716 was compared with a standard antibiotic treatment, those women receiving the probiotic treatment showed an increased improvement and lower recurrence of mastitis than those treated with the antibiotic, without showing significant differences between both treatments when considering their beneficial impact on breastmilk microbiota composition [139] (Table 5). Therefore, it can be concluded that the use of this strain constitutes an attractive strategy in the management of mastitis in women, which can constitute an alternative to the current antibiotic usage, thus providing evident additional benefits to the community and health system.

Viral infections

Viral infections such as influenza are a global health problem. Influenza is an acute highly contagious viral respiratory infection caused by the influenza virus. Infected patients can show a variety of symptoms including high fever, sore throat, headache, and running or stuffy nose, which can sometimes develop into pneumonia. At present, the most efficient strategy to control influenza is to provide protective adaptive immunity through the administration of a vaccine, although in some cases, the vaccine seems to show limited clinical effectiveness [140]. To improve it, the coadministration of the inactivated virus with adjuvants such as cholera toxin or heat-labile enterotoxin has been used [141] and a trace amount of cholera whole toxin as an adjuvant for nasal influenza vaccine [142,143]. However, the combination of the vaccine with these coadjuvants may have clinical risks [144]. In this sense, it has been postulated that *L. fermentum* CECT5716 can act as an adjuvant therapy to vaccines in preventing these viral infections. In fact, it has been reported that the oral administration of *L. fermentum* CECT5716 to adult volunteers receiving the anti-influenza vaccine, increased the Th1 response and virus-neutralizing antibodies, as well as increased the proportion of NK cells and the expression of *TNF-α* and *IL-12* after vaccination in these subjects, thus reinforcing the effect of the vaccine and potentially enhancing systemic protection from this viral infection [140] (Table 5).

Pediatric infections

Infectious diseases are the most common type of illness in infants worldwide. Many studies indicate that breast-fed children have a lower incidence of infections than formula-fed children, which could be associated with the modulation of gut microbiota by breast milk components [145]. Indeed, many studies suggest that the gut microbial profile of breastfed infants is dominated by *Bifidobacterium* [146,147], with the addition of a few other anaerobes and small numbers of facultative anaerobic bacteria [148], whereas in formula-fed children, the microbiota profile is different to infants fed with breast milk at all ages, but it is most apparent at 6 months of age. Moreover, breast milk-fed infants have a higher relative abundance of *Bifidobacterium* and lower abundance of bacteria of the *Clostridiaceae* family [149]. In fact, breast milk influences health-promoting microorganisms by factors such as polymeric IgA (pIgA), antibacterial peptides, and components of the innate immune response [150]. Moreover, and compared with formulas, breast milk has superior effects on the barrier integrity and mucosal defenses of the intestinal tract in children [151]. Because of its many benefits, exclusive breast-feeding for the first 6 months of life is recommended; however, and unfortunately, breast milk feeding to children is not always possible. It is interesting to note that although the composition of commercial formulas is more and more close to that of breast milk, the gut microbiota of breast-fed and formula-fed babies remains distinct [152]. For this reason, the manipulation of the intestinal microbiota of infants through the administration of probiotics can be considered as an interesting strategy for the prevention of community-acquired infections in children. Thus, the impact of an infant formula supplemented with *L. fermentum* CECT5716 was evaluated in a randomized controlled trial with the aim to determine the incidence rate of gastrointestinal infection in infants aged between 1 and 6 months [153]. The results revealed that the gastrointestinal infections were three times more frequent in infants of the control group than in those receiving the probiotic [153] (Table 5). In another randomized double-blinded controlled clinical trial, a formula supplemented with *L. fermentum* CECT5716 was administered to infants aged 6 and 12 months in order to evaluate its effect on the incidence of gastrointestinal and respiratory infections. The intake of the probiotic supplemented diet significantly reduced the incidence rate of gastrointestinal infections (46%) and of upper respiratory tract (27%) infections when compared to the corresponding control [28] (Table 5). More recently, the incidence of infections in infants has been also evaluated in a randomized double-blinded placebo-controlled multicenter trial in which the probiotic *L. fermentum* CECT5716 was administered to the mothers for 16 weeks. Almost 300 mother–infant pairs were included in this study. The results showed a significant lower incidence of conjunctivitis in the infants whose mothers received the probiotic. In addition, the group receiving the probiotic showed a lower load of *Staphylococcus* in the breast milk and in infant feces, which was associated with a significantly lower risk of respiratory and gastrointestinal infections. Therefore, this study showed that the treatment with this probiotic might neutralize the negative effect of the *Staphylococcus* load on respiratory and gastrointestinal infections in infants [28].

## 3. Concluding Remarks

The different preclinical and clinical studies performed with *Limosilactobacillus fermentum* CECT5716 have evidenced its anti-inflammatory and immunomodulatory properties and elucidated the precise mechanisms of action involved in its beneficial effects. This would justify its potential as a nutritional supplement in different inflammatory-related conditions, both for treatment and prevention. Of note, the properties reported for this probiotic cannot be generalized to other microorganisms, neither from the same genus nor from other strains of the same species. However, and despite the promising prospect of *L. fermentum* CECT5716, corroborative studies are needed for further understanding the role of this probiotic in the modulation of gut microbial populations, which seems a key mechanism associated with its immunomodulatory and anti-inflammatory effects both in healthy and disease.

## Figures and Tables

**Figure 1 nutrients-13-01016-f001:**
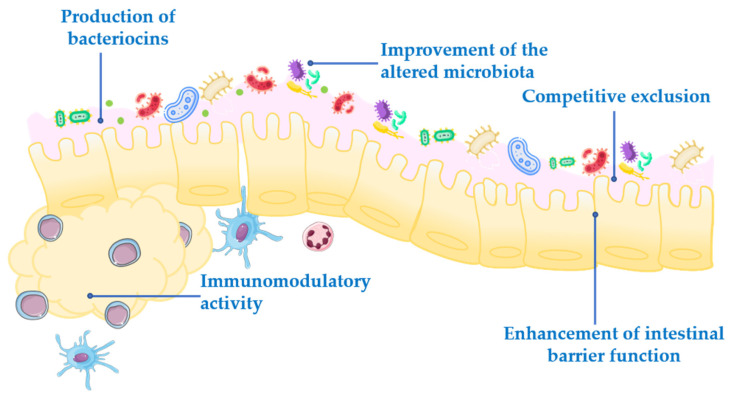
Mechanism of action of probiotics.

**Table 1 nutrients-13-01016-t001:** Effects of different *L. fermentum* strains in preclinical studies.

Strain/Origin	Properties	Mechanism of Action	Model	Reference
*L. fermentum* UCO-979C/human gastric tissue	Anti-inflammatory effect	↑ IL-10 production	THP-1 cell line	[42]
*L. fermentum* SRK414/ unknown (Food Microbiology Laboratory, Korea University (Seoul, South Korea))		↓ *TNF-α* and *IL-1β* expression in LPS-stimulated inflamed	HT-29 cell line	[43]
*L. fermentum* KBL374 and *L. fermentum* KBL375/feces of healthy Koreans		↓ Leukocyte infiltration and Disease Activity Index	DSS-induced colitis in C57BL/6N mice	[25]
*L. fermentum* MG901 and *L. fermentum* MG989/healthy woman’s vagina	Urogenital and intestinal anti-infective activity	Inhibit the *Candida albicans* growth	HT-29 cell line	[44]
*L. fermentum* 3872/milk of healthy women		Inhibition of *Campylobacter jejuni* growth and attachment to collagen I	Collagen I coated plates	[37]
*L. fermentum* L23/vaginal smears of healthy woman		Growth inhibition of *Gardnerella vaginalis*	Vaginal infected BALB/C mice with *Gardnerella vaginalis*	[45]
*L. fermentum* LF31/unknown (Milonet, Bromatech s.r.l., Milan, Italy)	Antioxidant capacity	Antioxidant capacity detectable with the oxigenic radical absorbance capacity (ORAC) assay	HT-29 cell line	[46]
*L. fermentum* JX306/Chinese traditional fermented vegetable		↓ Malondialdehyde (MDA) levels ↑ Activity of glutathione peroxidase (GSH-Px)	D-galactose-induced aging KM mice	[47]

Table symbols: ↓ Reduction and ↑ increase.

**Table 2 nutrients-13-01016-t002:** Effects of different *L. fermentum* strains in clinical trails.

Strain/Origin	Main Properties	Physiological Conditions	References
*L. fermentum* RC-14/female urogenital tract	−37% of improvement in vaginal flora - ↑ lactobacilli population - ↓ yeast levels	Healthy women	[48]
*L. fermentum* VRI-003 PCC/unknown (^®^Probiomics Ltd., Eveleigh, NSW, Australia)	- ↑ lactobacilli population - ↓ in the severity of gastrointestinal and respiratory illness in sportive males.	Competitive athletes	[49]
	- ↓ Severity Scoring of Atopic Dermatitis (SCORAD) index. - ↑ the cases of mild atopic dermatitis.	Children with atopic dermatitis	[50]
Combination of *L. fermentum* LF10 and *L. acidophilus* La02/vaginal swabs of healthy women or from direct brushing of gut mucosa of healthy humans	↓ 72% of clinical recurrences vaginal infections of vulvovaginal candidiasis	Women with recurrent vulvovaginal candidiasis	[51]
Combination of *L. fermentum* LF15 and *L. plantarum* LP01/feces of healthy humans or vaginal swabs of healthy female subjects	Reduction in the Nugent score and restoration (58%) of the vaginal microbiota of women	Women diagnosed with bacterial vaginosis	[52]
*L. fermentum* ME-3/healthy Human intestinal tract	- ↑ lactobacilli in feces -Improvement the blood Total Antioxidative Activity and Total Antioxidative Status	Healthy humans	[53]
Combination of *L. fermentum* ME-3 and a food supplement/healthy human intestinal tract	-Improvement cardiovascular and diabetes risk profile. - ↓ total cholesterol	Clinically asymptomatic humans	[54]
Combination of *L. fermentum* ME-3, *L. paracasei* 8700:2 and *Bifidobacterium longum* 46/healthy human intestinal tract and human feces	- ↑ the blood Total Antioxidative Status - ↓ oxidized/reduced glutathione ratio	Adult volunteers without gastric symptoms	[55]
Combination of *L. fermentum* LN99, *L. gasseri* LN40, *L. casei subsp. rhamnosus* LN113 and *Pediococcus acidilactici* LN23)/vaginal flora of healthy women	-Vaginal colonization of lactobacilli - ↓ recurrences of bacterial vaginosis and vulvovaginal candidiasis	Women diagnosed and treated for vulvovaginal candidiasis and bacterial vaginosis	[56]

Table symbols: ↓ Reduction and ↑ increase.

**Table 3 nutrients-13-01016-t003:** In vitro studies: mechanisms of action of *L. fermentum* CECT5716.

Experimental Models	Mechanisms of Action	Cell Model	Reference
Epithelial cell lines	↓ Expression of pro-inflammatory profile (*Il-6*) and ↑ the mucins in stimulated cells	CMT-93	[58]
	Restoration of *miRNA-150*, *miRNA*-*155,* and *miRNA*-*375* expression		
	↓ NO, IL-8, and IL-1β in stimulated cells	Caco-2	[65]
	↓ MAPK p42/44 ERK and p38 in stimulated cells		
Immune cells	↓ Pro-inflammatory mediators of stimulated cells (TNF-ɑ and IL-1β) and ↑ anti-inflammatory mediators (IL-10)	BMDM	[71]
	Restoration of *miRNA-150*, *miRNA-155,* and *miRNA-375* expression		[58]
	↓ IL-1β and NO production in stimulated cells	RAW-264.7	[65]
	Enhanced immune responses: Induction of the production of cytokines (TNFα, IL-1β, IL-8, MIP-1α, MIP-1β, and GM-CSF), activation of NK and T cell subsets, expansion of Treg cells	PBMC	[83]

**Table 4 nutrients-13-01016-t004:** Mechanisms of action of *L. fermentum* CECT5716 in animal models.

Experimental Models	Mechanisms of Action	Animal Model	Reference
Experimental colitis	↓ Immune response: *- Tnf-**ɑ, iNos,* and *Il-6* expression. - LTB_4_ and TNF-ɑ protein levels. - MPO activity.	Rat	[65,95]
	↑ Antioxidant activity: GSH content.		
	Induced growth of Lactobacilli species and increased more than doubled the production of the SCFAs (acetate, butyrate, and propionate)		
	Amelioration of the weight decrease in a 20% and amelioration of diarrhea incidence and gut dysbiosis	Mouse	[27,58]
	↓ *Tnf-**ɑ, Il-1β*, *iNOS* and *Mmp-9* expression		
	Restoration of *miR-155* and *miR-223* expression		
	Microbiota restoration: increase microbial diversity and restore the F/B ratio.		
Metabolic syndrome	Prevent liver steatosis and inflammatory status	Rat	[112]
	↓ Glucose and insulin levels in plasma		
	Gut dysbiosis restoration by preventing the increase in *Bacteroidetes* and the reduction in *Firmicutes.* Increase the levels of *Akkermansia muciniphila*		
	Prevention of hypertriglyceridemia and hyperleptinemia		
	↓ Body weight gain in 15–20%	Mouse	[114]
	Amelioration of glucose and lipid metabolism		
	↑ *Glut*-4 expression		
	↓ *Tnf-α* and *Il-1β* expression and inhibition of NADPH activity in aortic tissue		
	Restoration of impaired endothelial disfunction		
	Anti-inflammatory properties: ↓ *Il-6, Tnf-α, Mcp-1,* and *Jnk-1* expression in liver and fat		
	Amelioration of obesity-associated dysbiosis: - ↑ Richness and diversity - Restore F/B ratio, decreasing it. - Restore levels of *Verrumicrobia,* *Akkermansia* and *Bacteroides.* *-* ↑ Lactate- and acetate-producing genera.		
	Enhanced intestinal epithelial integrity (↑ occludin levels) and ↓ LPS plasma level		
Systemic lupus erythematosus	Prevention gut dysbiosis: - Restore F/B ratio. - ↑ *Bifidobacterium* and *Parabacteroides* genera. - ↓ *Blautia* and *Lachnospira.*	Mouse	[57]
	↓ Pro-inflammatory cytokine (*Tnf-α* and *Il-1β* expression) /plasma levels of LPS		
	Intestinal integrity amelioration (↑ *Zo-1* and *Occludin* expression)		
	↓ Hypertension		
	Prevention of the endothelial dysfunction (↑ acetylcholine-induced vasodilation)		
	↓ NAPDH oxidase activity		
	Prevention of the altered T-cell polarization		
Pregnancy and lactation stage	↓ Cytotoxic T cells	Rat	[135]
	↑ γ-linolenic acid ↓ Saturated fatty acids		

**Table 5 nutrients-13-01016-t005:** Clinical studies: mechanisms of action of *Limosilactobacillus fermentum* CECT5716.

Disease	Mechanism of Action	References
Mastitis	Prevention of lactational mastitis symptoms	[143,144]
	↓ IL-8 levels	
	↓ *Staphylococcus* load in the breast milk	
Pediatric infections	↓ Incidence of gastrointestinal and respiratory infection	[28,153]
	↓ Incidence of conjunctivitis and the load of *Staphylococcus*	[28]
Vaccination stage	↑ Th1, NK cells, and *Tnf-α* and *Il-12* expression	[145]

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
