# Peer review of "Limosilactobacillus fermentum CECT5716: Mechanisms and Therapeutic Insights"

_nutrients, 2021, doi:10.3390/nu13031016_

Round 1

Reviewer 1 Report

Ms. ID nutrients-1111847: Limosilactobacillus fermentum CECT5716: mechanisms and therapeutic
insights

General comments:

Rodríguez-Sojo and collaborators submitted a synthesis of the work carried out on Limosilactobacillus fermentum strain CECT5716 (Lc40) in pre- and clinical studies. The first part describes the main studies based on cellular or experimental models and ends with a synthesis of the clinical study results.

This type of literature review is generally useful to document a probiotic strain and thus allow the scientific community to know its health benefits and its probiotic potential.

Specific comments:

This literature review submitted by Rodríguez-Sojo and collaborators has been carefully written both in the presentation of the literature data and the presentation of the figures.

Minor comments:

Title : suggestion please indicate Lc40

 Limosilactobacillus fermentum CECT5716 Lc40 : mechanisms and therapeutic insights.

For example

Pastor-Villaescusa, B., Hurtado, J. A., Gil-Campos, M., Uberos, J., Maldonado-Lobón, J. A., Díaz-Ropero, M. P., ... & Olivares, M. (2020). Effects of Lactobacillus fermentum CECT5716 Lc40 on infant growth and health: a randomised clinical trial in nursing women. Beneficial microbes11(3), 235-244.

The quality of the references is as important as that of the manuscript and denotes the particular attention of the authors to submit a manuscript without error.

Reference section filled with mistakes (lack of citation accuracy (ref: 129 );  incorrectness of abbreviations and punctuation of journal names (ref: 129);  omission to italicize the scientific names of bacteria (ref: 5, 14-20, 25-30, 32-34, 36-37, 39, 41, 55, 59, 64, 71, 83, 85, 96-99, 101, 120, 123, 133-134); article titles should be in sentence style, without the upper case letters throughout (ref: 5, 8, 10, 16, 29, 32, 37, 47, 48, 50, 60, 65, 78, 85, 90, 91, 94, 95, 98, 101, 105, 114, 116, 120, 122, 123, 132 ).

Reviewer 2 Report

In their manuscript, Rodriguez-Sojo et al. described the properties of a single probiotic strain, Limosilactobacillus fermentum CECT5716, previously Lactabacillus fermentum. The review provides very detailed information on the studies regarding CECT5716 probiotic effects. This is an asset for scientists interested by this specific probiotic but may lack general interest for people working in the field of probiotics but not specifically on that given strain. This is why I recommend to add a bit more information on other L. fermentum probiotic strains before focusing on CECT5716.

  1. General comment you are using both LC40 and CECT5716; only use the current name, fermentum CECT5716, in the core of the review.
  2. Line 94: generally without catalase activity (see for instance Engesser et al 1994)
  3. Line 115: ref 25 does not seem appropriate as it only concerned antibacterial activities
  4. Line 136 its mechanisms of action
  5. Adding information on other fermentum probiotics strains will increase the general interest of this review. Please add a table : strain number, origin and main claims/probiotic properties; whether the strain has been tested in clinical trials
  6. Reorganize the manuscript to start with what have been established in vitro using cell lines, then what has been confirmed or not in the various preclinical animal models and finally clinical trials.
  7. Put more details in tables and less in the core text. For instance,
    • Indicate the effective per group in preclinical or clinical trials
    • “â tnf-a, iNos, LTB4, IL6, MPO activity and GSH » clarify what concerns gene expression or activity, what is related to immune or oxidative stress responses
    • “induced growth of lactobacilli species and production of SCFA” : quantify if possible
    • Replace “Decrease pro-inflammatory mediators of stimulated cells” by information on the observed effect: which mediators?
    • “gut dysbiosis restoration”: replace this general comment by information on what has been observed
    • “decrease body weight”: quantify the effect (5%, 20%..)
    • etc
  8. Table 1: group together cellular models and separate them from animal models
  9. Lines 609-618: not clear why the adjuvant therapy to vaccines is in the section of pediatric infections. I suggest to make another short section excepted if the adjuvant effect was tested on pediatric vaccines, in this case it should be stated.

Round 2

Reviewer 2 Report

I appreciated the quality work done by the authors to address the comments. Well done!